# A Thermostable Lipase Isolated from *Brevibacillus thermoruber* Strain 7 Degrades Ɛ-Polycaprolactone

**DOI:** 10.3390/biotech12010023

**Published:** 2023-03-14

**Authors:** Nikolina Atanasova, Tsvetelina Paunova-Krasteva, Margarita Kambourova, Ivanka Boyadzhieva

**Affiliations:** Institute of Microbiology, Bulgarian Academy of Sciences, Acad. G. Bonchev Str., Bl. 26, 1113 Sofia, Bulgaria; nikolina@microbio.bas.bg (N.A.); pauny@abv.bg (T.P.-K.); margikam@microbio.bas.bg (M.K.)

**Keywords:** thermostable lipase, enzyme decomposition of plastics, Ɛ-polycaprolactone biodegradation, polycaprolactone biodegradation products

## Abstract

The tremendous problem with plastic waste accumulation has determined an interest in biodegradation by effective degraders and their enzymes, such as thermophilic enzymes, which are characterized by high catalytic rates, thermostability, and optimum temperatures close to the melting points of some plastics. In the present work, we report on the ability of a thermophilic lipase, by *Brevibacillus thermoruber* strain 7, to degrade Ɛ-polycaprolactone (PCL), as well as the enzyme purification, the characterization of its physicochemical properties, the product degradation, and its disruptive effect on the PCL surface. The pure enzyme showed the highest reported optimum temperature at 55 °C and a pH of 7.5, while its half-life at 60 °C was more than five hours. Its substrate specificity referred the enzyme to the subgroup of lipases in the esterase group. A strong inhibitory effect was observed by detergents, inhibitors, and Fe^3+^ while Ca^2+^ enhanced its activity. The monomer Ɛ-caprolactone was a main product of the enzyme degradation. Similar elution profiles of the products received after treatment with ultra-concentrate and pure enzyme were observed. The significant changes in PCL appearance comprising the formation of shallower or deeper in-folds were observed after a week of incubation. The valuable enzyme properties of the lipase from *Brevibacillus thermoruber* strain 7, which caused a comparatively quick degradation of PCL, suggests further possible exploration of the enzyme for effective and environment-friendly degradation of PCL wastes in the area of thermal basins, or in thermophilic remediation processes.

## 1. Introduction

Several decades ago, synthetic polymers with long molecules, named plastics, were introduced as substitutes for natural materials and since then they have invasively entered almost every area of industry and social life by making them more colorful, easier, and safer [1]. Despite the undoubted impact of plastics in erasing the human life standard, the exponential accumulation of their non-degradable products in the environment represents a growing concern for wildlife and provokes enormous interest in searching for degradable products, or new alternatives to reduce waste. Resistance to degradation by the pressure of environmental physicochemical factors has attracted scientific and technological interest in the process of degradation by microorganisms and their enzymes. However, a common feature of the characterized plastic-degrading enzymes is their low activity due to the short period of plastic presence in nature and the correspondingly short time of enzyme evolution [2]. Human society is starved of developing highly effective processes for managing the negative impact of plastic pollution.

Among the widely used plastics, poly-Ɛ-caprolactone (C_6_H_10_O_2_)n (PCL) is produced from crude oil by the ring-opening polymerization of Ɛ-caprolactone and 2-methylene-1-3-dioxepane has attractive polymer characteristics, such as low-temperature flexibility, gloss, UV- and wear resistance, non-hazardous nature, and good biocompatibility, which determines its applicability in implants, surgical drapes, wound dressing, injection-molded devices, drug delivery, etc. [3]. Polyester is the preferred plastic for making automotive components, general-purpose tubing, construction materials, models, packaging, jewelry, prototyping, dental splints, and root canal filling; however, its market price in USD is ~10/kg [4], which is not attractive enough to enlarge its consumption.

There are very few reports on the biological degradation of PCL by microorganisms, including by members of the filamentous fungi from the genera *Penicillium*, *Aspergillus,* and *Rhizobium*, which are able to grow on a mineral medium containing this polymer [5,6], although their enzymes have not been characterized. Moreover, several mesophilic bacteria have been reported as capable of degrading PCL, such as *Bacillus subtilis* [7], *Bacillus sp*. [8], *Pseudomonas*, *Alcanivorax*, and *Tenacibaculum* [9]. Isolates from the genera *Pseudomonas* and *Rhodococcus* degraded 53% (*w*/*w*) of the PCL films during 30 days of incubation [10].

Extremophilic microorganisms and relevant extremophilic enzymes appear to be a promising solution to the plastic accumulation problem due to the changes that occur in polymer properties at extreme values of temperature, pH, and salinity. Among the few reports available for thermophilic plastic degraders are polyethylene degradation by *Brevibaccillus borstelensis* [11], polyethylene terephthalate by *Clostridium thermocellum* [12], and nylon degradation by *Anoxybacillus rupiensis* and *Geobacillus thermocatenulatus* [13,14]. Thermophilic enzymes able to degrade PCL could be especially useful in plastic degradation due to its low melting point of around 60 °C. An efficient PCL degradation by the representatives of *Streptomyces* has been reported at 40–50 °C [15,16]; however, their enzymes have not been isolated.

Enzymes able to degrade plastics belong to two main classes of enzymes, the hydrolases and oxidases, which comprise the esterase, protease, cutinase, dehydrogenase, or laccase enzymes [17]. Investigations on the best-studied enzymes isolated mainly from mesophilic organisms such as polyurethane (PUR)- and polyethylene terephthalate (PET)ases, which are active on some of the high-molecular-weight synthetic polymers, have revealed that they still have moderate turnover rates; however, whether the enzymes degrade some plastics, such as polystyrene (PS), polyvinyl chloride (PVC), polypropylene (PP), and polyamide is not known [18]. Esterase and particularly lipase activities have been detected in strains that are able to degrade many plastics, including the polyester PCL [19,20]. The only reported and partially purified PCL degrading enzyme has been synthesized using the mesophilic bacterium *Alkaligenes faecalis* TS22 [21].

Our previous work provided evidence for effective PCL degradation by the thermophilic bacterium *Brevibacillus thermoruber* strain 7 and revealed the community from which the strain was isolated [1]. In the present paper, we report on the isolation of lipase from this strain, its purification, the characteristics of the purified enzyme, the identification of the degradation products, and changes to the PCL surface by the enzyme.

## 2. Materials and Methods

### 2.1. Microorganism

*B. thermoruber* strain 7, active on Ɛ-polycaprolactone, was isolated from a hot spring in Marikostinovo, located in Southwest Bulgaria with a water temperature of 55 °C and pH of 7.8 [1].

### 2.2. Media and Culture Conditions

PCL pearls with a diameter ranging from 2.9 to 4.8 mm and an average MW 80,000 (Sigma-Aldrich, Steincheim am Albuch, Germany) were used as the only carbon source. Before adding to the basal medium, PCL pearls were soaked in 96% ethanol followed by five minutes of sonication at room temperature for sterilization and washed with sterile distilled water. The microbial cells were grown in 1500 mL basal medium containing: NH_4_NO_3_, 0.01%; KH_2_PO_4_, 0.03%; K_2_HPO_4_, 0.14%; MgSO_4_, 0.01%; FeSO_4_·7H_2_O, 0.002%; Na_2_MoO_4_·2H_2_O, 0.0005; yeast extract, 0.1%, pH 7.5. PCL pearls were added to the basal medium at a final concentration of 0.3 ± 0.02%. The cultivation was performed for 48 h at 55 °C, 80 rpm of the shaker.

### 2.3. Enzyme Assay

Esterase assay was used for an estimation of the enzyme activity in the supernatant as described previously [22]. Hydrolysis of p-nitrophenyl fatty acid as a substrate was determined spectrophotometrically after one hour of incubation at 55 °C in 0.02 M Tris-HCl buffer, pH 7.5 at 405 nm. One unit of esterase activity was determined as the amount of enzyme needed to liberate 1 μM p-nitrophenol per minute in the described conditions. A molar extinction coefficient for p-nitrophenol at pH 7.5 was found to be 3.39 × 10^3^/M.

The previously described method for screening PCL active microorganisms [23] with some modifications was used in the current work as a quality enzyme test. PCL (100 mg), with an average MW 14,000 (Sigma-Aldrich, Steincheim am Albuch, Germany), was solved in 10 mL acetone on a magnetic stirrer at 50 °C. The emulsion was added through a sterile filter to the sterile Tris-HCl buffer supplemented with 1.5% agar and 0.01% NaN_3_. The magnetic stirring continued until acetone evaporated and the medium was poured into plates. After cooling 50 μL from each, the pure enzyme (PE), ultraconcentrate (UC), and a buffer as the control were dropped into the holes. PE and UC were pre-leveled to contain 2900 U/mL each. Biodegradation activity was determined by the formation of clear halos around the holes after 24 h incubation of Petri dishes at 55 °C.

Protein was determined quantitatively by the method of Lowry et al. [24] with bovine serum albumin as a standard.

All data are averages from at least three analyses.

### 2.4. Enzyme Purification

The ultraconcentrate was received after centrifugation of the culture liquid (1500 mL) at 4000× *g* for 15 min and 300-fold concentration using Millipore standard ultrafiltration cell (500 mL) with Mw 30 kDa cut-off membrane (Millipore Corporation, Billerica, MA, USA). The concentrated protein was passed over a DEAE Sepharose^TM^ Fast Flow (GE Healthcare Bio-Sciences AB, Upsala, Sweden) column (1.6 × 60 cm) with a bed volume of 68 mL. The column was previously equilibrated with 20 mM Tris buffer, pH 7.5. After washing, the proteins were eluted with buffer followed by a step NaCl gradient 0.12 M, 0.18 M, 0.25 M, 0.5 M, and 1.0 M. All fractions were tested for enzyme activity. The active fractions were pooled, concentrated on 30 kDa and further used for electrophoresis.

Sodium Dodecyl Sulfate Polyacrylamide Gel Electrophoresis (SDS-PAGE) was performed according to Laemmli [25] in 10% polyacrylamide resolving gel. Marker Rule (Thermo Scientific™ PageRuler™ Plus Prestained Protein Ladder, Rockford, IL, USA) containing proteins with a molecular mass 10-250 kDa was used for an estimation of the enzyme mass.

### 2.5. Gel Permeation Chromatography

One mL of PE or UC (2900 EU for each) was incubated with one PCL pearl for each sample with a gravimetric weight 18 mg ± 3% (MW 80,000) for a week at 55 °C. A pearl in buffer was used as a control. Each sample was triple-repeated. After a week the samples were frozen in a 30 mL glass vial at −15 °C and kept in a lyophilic dryer at 60 °C with a 80 mTor vacuum for 12 h and dissolved in 1 mL HPLC grade tetrahydrofuran (THF) for 2 h at 45 °C. The samples were filtered through a 0.45 μm PTFE syringe filter in standard salinized 2 mL chromatography vials. A GPC analyzer (Model SHIMADZU Nexera) equipped with 5-Channel Degasser DGU-20A, HPLC Pump LC-20AD, Autosampler SIL-20AC, Column Oven CTO-20AC, and Refractive Index Detector RID-20A was operated at 45 °C and a THF elution rate of 1.0 mL/min. The used GPC Column Set was PSS SDV 50 Å (300 mm × 8.00 mm × 5 μm), PSS SDV 100 Å (300 mm × 8.00 mm × 5 μm), and PSS SDV Linear M (300 mm × 8.00 mm × 5 μm). Instrument Control, Data Acquisition and Data Processing were performed by GPC software Lab Solutions v.5.71 SP1.

### 2.6. Characterization of the Purified Lipase Properties

The optimal temperature and pH for an enzyme action of UC and PE were determined after incubation of the reaction mixture at different temperatures in the range of 40–80 °C with 5 °C steps, pH 7.5. The optimum pH was determined by resolving the substrate in 0.2 M buffer (acetate, phosphate, or glycine-NaOH) in the pH range 6.0-9.0 with 0.5 steps, temperature of 55 °C. The thermostability was determined by pre-incubation of the lipase for different times in Tris-HCl buffer (pH 7.5) at 60 °C in the absence or presence of 5 mM CaCl_2_ and the residual activity was determined at 55 °C. The enzymatic activity without pre-incubation was denoted as 100%.

The effect of metal ions, EDTA, surfactants, and inhibitors on the enzyme activity was investigated by amending the samples for 30 min at room temperature with the selected chemicals (5 mM): ZnSO_4_·7H_2_O, KCl, CoCl_2_·6H_2_O, NaCl, CaCl_2,_ FeCl_3_·6H_2_O, MnCl_2_·4H_2_O, MgCl_2_ ·6H_2_O, HgCl_2,_ CuSO_4_·5H_2_O, sodium dodecyl sulfate (SDS), Tween 20, ethylenediamine tetraacetic acid (EDTA) disodium salt, phenylmethylsulfonyl fluoride (PMSF), dithiothreitol (DTT), Na-lauryl sulfate (NaLS), and N-bromsuccinimide (NBS). The residual lipase activity was determined after an assay at optimum conditions against a control that did not contain added chemicals.

### 2.7. Scanning Electron Microscopy (SEM)

One PCL pearl with a gravimetric weight of 18 mg ±3% (MW 80,000) was incubated with 0.5 mL PE (1450 EU), UC (1450 EU), or buffer for a week at 55 °C. Each sample was repeated at three Eppendorfs. The pearls were fixed for 2 h in 4% glutaraldehyde in 0.1M Na cacodylate buffer (pH 7.2), then washed and post-fixated in 1% OsO_4_ for 1 h. Dehydration was performed in graded ethanol series. After being sputter-coated with gold using Edwards’s sputter coater, the samples were examined by SEM (Philips scanning electron microscope, Phillips, Amsterdam, The Netherlands), at an accelerating voltage of 30 kV.

## 3. Results

### 3.1. Enzyme Purification

The enzyme synthesized by *B. thermoruber* strain 7 was purified in a three-step scheme (Table 1) with a purification factor of 23.1-fold and a final yield of 51.4%. The critical loss in the step of ultraconcentration was probably connected with the clumping of lipase molecules in the water system under the pressure due to their lipophilic nature. The peak that appeared after elution with buffer was not active, neither were the peaks at 0.12 M, 0.18 M, 0.5 M, and 1.0 M NaCl; an active one appeared after supplying of 0.25 M NaCl. The PCL-degrading enzyme was determined to be homogeneous as indicated by SDS-PAGE. The molecular mass of the purified lipase estimated by SDS-PAGE was 28 kDa (Figure 1).

The qualitative test demonstrated the presence of UC and PE activity after incubation in PCL-containing Petri dishes (Figure 2).

### 3.2. Lipase Properties

The determination of the substrate specificity of the enzyme by pNPP or pNPB clearly demonstrated its lipase nature (Figure 3). The established activity of the enzyme fluently increased up to 280 U/mL at 55 °C with pNPP and lower values were observed for all temperatures with pNPB reaching up to 116 U/mL at 50 °C. Measuring the activity at temperatures higher than 50 °C with pNPB was not possible due to the self-destruction of pNPB at that temperature.

The lipase from strain 7 had a typical forthe thermophilic enzymes temperature range of activity—from 45 °C to 65 °C with an optimum at 55 °C, 80% relative activity at 60 °C and 40% at 65 °C. Its activity was not less than 80% in a very wide pH area from 6.0 to 9.0 with an optimum at 7.0–8.0. The enzyme from *B. thermoruber* strain 7 had a half-life of 5 h at 60 °C and increased up to 6 h in the presence of 5 mM CaCl_2_ (Figure 4).

The enzyme activity was inhibited by most of the metal ions, especially strong Fe^3+^ influence (Table 2). The only metal ion with a positive effect on the activity was Ca^2+^. The chelating agent EDTA fully inhibited the enzyme. The enzyme was sensitive to detergents and inhibitors. The strong inhibition by the tryptophan inhibitor NBS, the thiol inhibitor PCMB, the serine inhibitor PMSF, and NaLS was observed.

### 3.3. Biodegradation Products as a Result of the Enzyme Action

The comparison of the penetration chromatography profile of PCL intermediates revealed an almost identical elution pattern when PCL was degraded by UC or by PE that indicated the isolated lipase as the only active PCL enzyme synthesized by strain 7 (Figure 5). The elution volume of the main peak in both samples was very similar, 65.0% and 66.9% of the whole area for PE and UC, correspondingly, and it appeared at 28.0 mL for the sample of PE and 28.4 mL for UC. An estimated molecular weight of 115 and 125 for PE and UC, correspondingly is similar to the molecular weight of the monomer Ɛ-caprolactone (114). A peak (13.6% of the whole area) with an elution volume of 26.5 mL and MW of 252 corresponding to a dimer was observed only in the PE sample. Second, the biggest peak with an elution volume of 25.6 mL and 25.7 mL with a peak area of 14.7% and 23.6% for PE and UC, correspondingly, and MW of 454 represented a tetramer. The elution volumes of the two smallest peaks in both samples corresponded to intermediates with a molecular weight of the octamer and dodecamer.

### 3.4. SEM Investigations of the Morphological Changes

Analysis by SEM morphological changes on the surfaces of the PCL pearls after their week of incubation in a buffer solution of UC and PE (3000 U/mL) revealed deep surface damages (Figure 6). The SEM images of PCL control cultivated in the absence of enzyme exhibited relief with no characteristic surface defects after the incubation period (A). Enzymatic digestion was confirmed by the observations of a plastic surface with added UC (B) and PE (C, D). Very similar indication effects of degradation with clearly distinguishable holes (asterisk) and grooves (arrows) in the plastic were observed for both samples that confirmed again the only PCL active enzyme synthesized by this strain was the studied lipase. The SEM micrograph after the higher magnification (D) revealed the biodegrading ability of the enzymes even in the lower layers of the PCL plastic pearls. A grainy structure material probably containing undecomposed plastic fragments was observed in both samples.

## 4. Discussion

Most of the best-studied and explored esterases, such as PUR-, PET-ases, as well as the PCL active enzyme from *Alcaligenes faecalis* have been synthesized from mesophiles. To the best of our knowledge, the isolated and characterized lipase from *B. thermoruber* strain 7 is the first reported thermostable enzyme able to degrade PCL and among the small number of thermophilic enzymes capable of degrading plastics. The scarcity of information on the enzyme characteristics, the degradation products, and the clearing of the biochemical mechanisms [26] hindered the comparison between the lipase from a strain 7 and other enzyme properties.

The observed recovery yield (51.4%) for strain 7 lipase after three-stage purification represented a comparatively good value in comparison with other lipases [27,28]. The established molecular weight of 28 kDa was similar to that for the enzyme from *Bacillus subtilis* active on polyurethane (28 kDa) [29] and for the recombinant polyesterase from *Thermobifida alba* expressed in *Escherichia coli* (30 kDa) [30]. The temperature optimum of this enzyme (55 °C) coincided with the melting point of 55–60 °C for PCL [31] making the thermophilic enzyme especially appropriate for effective PCL degradation. We also established experimentally that the melting temperature was 57 °C for MW 14 000 PCL and 60 °C for MW 80,000 (unpublished results). The chains in the amorphous polymer domains can gain enough mobility to access the active sites of the enzyme when the enzymatic hydrolytic reactions take place under a temperature close to the melting temperature [32]. Among thermophilic enzymes, PET-ase from the actinomycete *Thermobifida fusca* acting at 55 °C has decreased the initial weight by 50% after three weeks of incubation of commercial beverage bottles and films from the BASF [2]. An esterase from *Thermobifida alba* has been active on the aliphatic-aromatic copolyester film from 20 °C to 75 °C (with an optimal range of 45 to 55 °C) and in a pH range of 5.5 to 7.0 (with an optimal pH of 6.0) [30]. The cutinase from *Thermobifida* has shown remarkable thermostability—the tested PUR foils have lost a significant part of their weight after incubation for 200 h at 70 °C [33]. The enzyme from *B. thermoruber* strain 7 had a half-life of 5 h at 60 °C and it increased up to 6 h in the presence of 5 mM Ca^2+^. PET-ases from *Thermobifida* species are also Ca^2+^-dependent, especially in terms of their thermal stability [34]. The positive effect of Ca^2+^ on both activity and stability could be found in most of the reports [35]. The observed negative effect of the chelating agent EDTA confirmed that the enzyme activity depended on the Ca^2+^ presence in the active site. The enzyme from strain 7 demonstrated a high sensitivity to Fe^3+^, detergents, and inhibitors. Inhibition of the enzyme activity by transition and heavy metal ions was suggested to be a result of the formation of a complex with the reactive groups of the enzymes, denaturing the enzyme protein [36,37]. The inhibitory effect of detergents such as SDS, Tween 20 and DTT was suggested to be a result of the decreased surface tension of aqueous systems and possible modification of the enzyme distribution between the lipid surface and the aqueous phase [36,38]. The strong inhibition by the tryptophan inhibitor NBS, the thiol inhibitor PCMB and the serine inhibitor PMSF indicated an important role of these amino acids in the catalytic mechanism and their possible modification in the active site under the inhibitor influence. The effect of NaLS was related to its preference to form van der Waals’ forces and hydrogen bonds with the enzyme interface [39]. Significantly less sensitivity has been observed for the enzyme from *B. subtilis* that has not been inhibited by PMSF, adenylmethylsulfonyl fluoride, and EDTA [29]. Investigations on the lipase properties demonstrated typical for this subgroup of thermophilic enzymes characteristics; however, the enzyme from *B. thermoruber* 7 is unique in its ability to degrade PCL.

The higher substrate preference of the enzyme synthesized by strain 7 to pNPP than to pNPB clearly demonstrated its affiliation to the subgroup of lipases (EC 3.1.1.3) rather than to the true esterases (EC 3.1.1.1). The partially purified enzyme from *Alcaligenes faecalis* was also described as a lipase [21]. *B. subtilis* enzyme has shown the highest substrate specificity to p-nitrophenyl-acetate [29]. An esterase nature was detected also for the polyurethane degrading hydrolase [40]. *Thermobifida alba* expressed in *Escherichia coli* has degraded aliphatic-aromatic copolyester with a preference for p-nitrophenyl acyl esters (C2 to C8), indicating that the enzyme is an esterase rather than a lipase [30]. The putative enzymes active on polyesters have been identified by a genome mining approach as cutinase in *Pseudomonas pseudoalcaligenes* (PpCutA) and lipase in *Pseudomonas pelagia* (PpelaLip) [18].

GPC analysis of the degradation products revealed that the enzyme from strain 7 attacked the main chain of PCL by exo-mechanism of action with a monomer as a main product. As the only peak observed in the supernatant after 48 h of strain cultivation was the monomer [1], the appearance of additional intermediates after a longer time of enzyme action could suggest a slow-running reverse reaction. Usually, objects of the enzyme attack in most of the reported works have been the chemical additives rather than the polymer and a low rate of enzyme turnover has been observed for the main chain attack. For example, lipases and esterases have been able to rapidly degrade the emulsified PUR, while their capability to degrade the solid polyester film, foam, and elastomer has been weak [33]. Most of the enzymes have demonstrated plastic-specific activity; however, exceptions have been reported. Well-known PETase derived from *Ideonella sakaiensis* 201-F6 (IsPETase) has been characterized by high substrate specificity [41]; however, the same enzyme isolated from marine-sponge associated *Streptomyces sp*. SM14 was reported to also express some PCL activity [23].

SEM analysis revealed deep changes in the PCL pearl surface after a week. A similar deeply destroyed PCL surface has been observed during the growth of strain 7 in a medium with PCL as the only carbon source [1]. Visible changes on the PCL surface have been seen after longer cultivation of some strains in PCL containing medium—14 days cultivation of *Bacillus subtilis* at 30 °C [7], 20 days cultivation at 40 °C of *Bacillus sp*. [8], a week cultivation of the genera *Pseudomonas*, *Alcanivorax*, and *Tenacibaculum* at 25 °C [9]. However, an effect on the plastics after direct incubation with enzymes isolated from these microorganisms has not been described.

## 5. Conclusions

An extracellular lipase active on Ɛ-polycaprolactone was isolated after cultivation of *Brevibacillus thermoruber* strain 7 in a medium with PCL as the only carbon source. This enzyme represents the first thermostable enzyme active on PCL that was isolated, purified and characterized in relation to its properties, the received hydrolysis products, and plastic surface damages. An effective scheme for lipase purification was suggested that provided a high purification factor and a comparatively good yield. The high-temperature optimum, high thermostability, and large pH area of activity determine the enzyme as very appropriate for remediation processes of PCL wastes at temperatures close to PCL melting points such as polluted thermal areas or in situ treatment of plastic wastes which will be needed in the further estimation of the cost of processing.

## Figures and Tables

**Figure 1 biotech-12-00023-f001:**
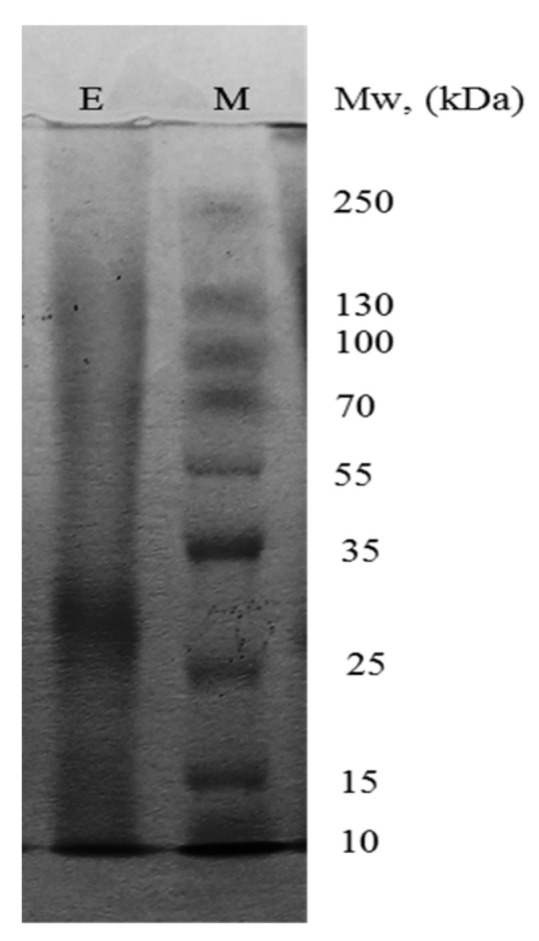
SDS-PAGE of the purified lipase from *Brevibacillus thermoruber* strain 7. E, purified enzyme; M, molecular mass markers.

**Figure 2 biotech-12-00023-f002:**
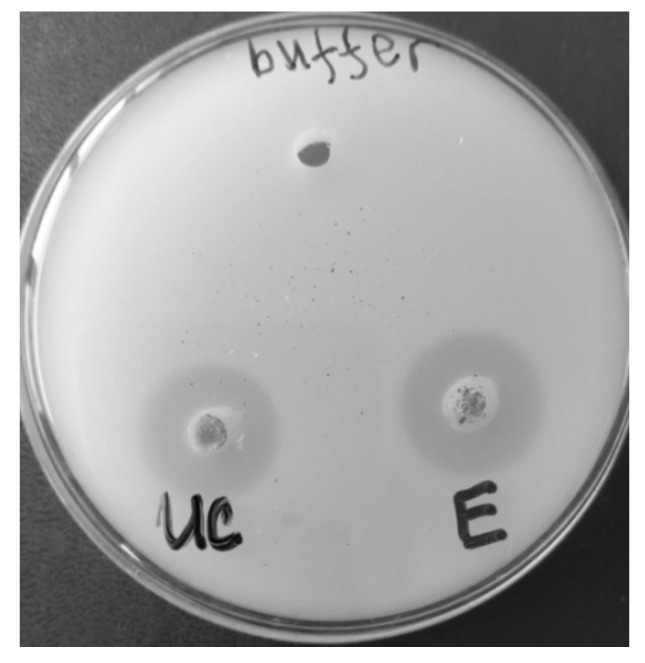
Clear halos were observed in agar plates containing 0.1% PCL (MW 14,000) when 50 μL containing 145 U of UC (ultraconcentrate) or E (pure enzyme) were incubated in the holes at 55 °C for 24 h. Buffer was used as a control.

**Figure 3 biotech-12-00023-f003:**
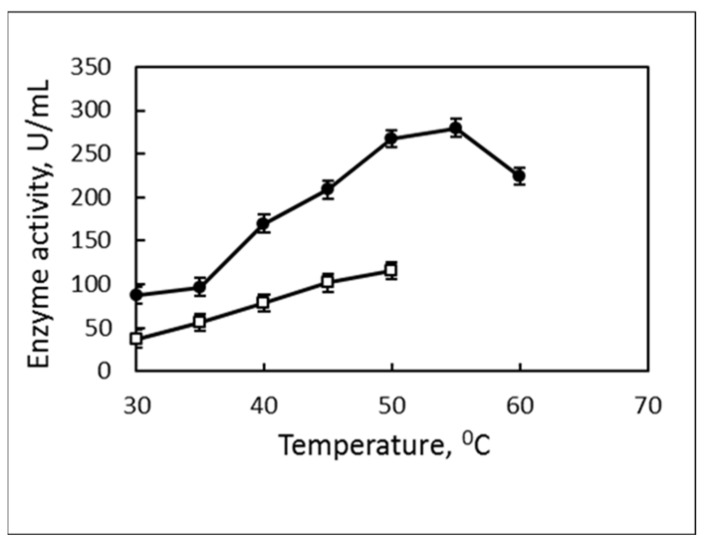
Substrate preference of the enzyme synthesized by a strain 7. □, pNPB; ●, pNPP.

**Figure 4 biotech-12-00023-f004:**
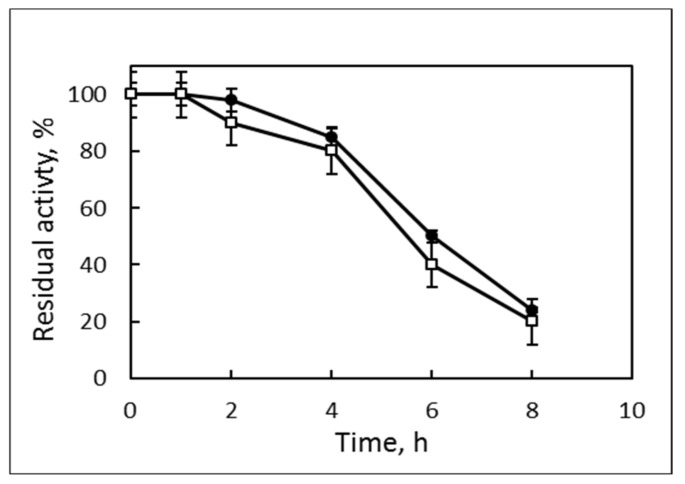
Thermostability of the purified lipase after preliminary incubation at 60 °C for different times (hours). □, PE; •, PE + 5mM CaCl_2_. The residual activity was determined at 55 °C.

**Figure 5 biotech-12-00023-f005:**
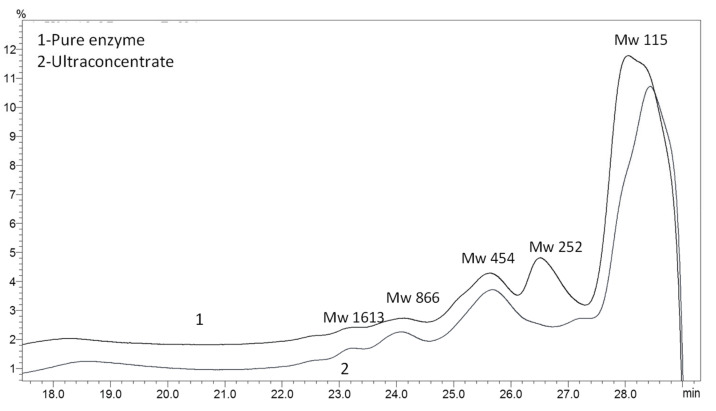
GPC elution pattern of PCL degradation intermediates after a week of cultivation of *B. thermoruber* strain 7. (1), PE; (2), UC.

**Figure 6 biotech-12-00023-f006:**
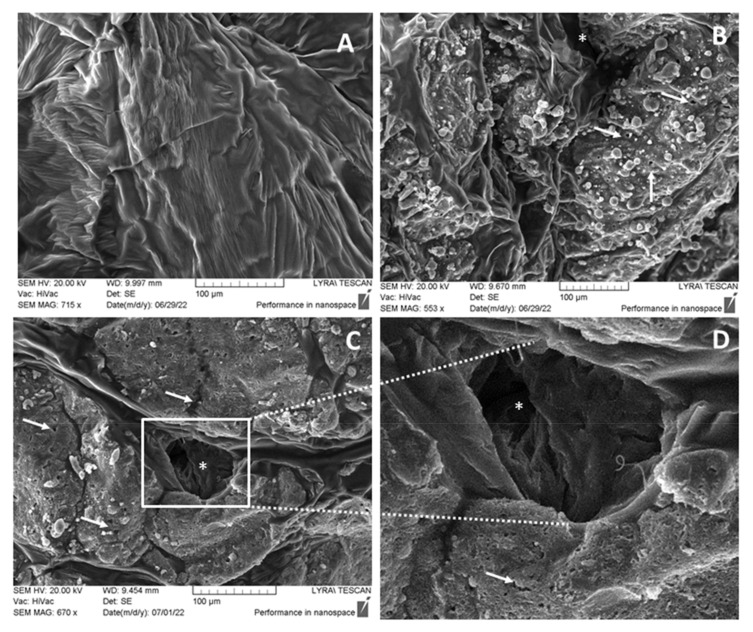
Scanning electron images of PCL pearls indicated degradation processes. (**A**) Control probe; (**B**) PCL pearls incubated in the presence of UC; (**C**,**D**) PCL pearls incubated in the presence of PE. The appearance of holes (asterisk) and grooves (arrows) in the plastic indicated the morphology changes during enzymatic degradation. Bar = 100 µm.

**Table 1 biotech-12-00023-t001:** Purification profile of the lipase from *Brevibacillus thermoruber* strain 7.

Purification Step	Volume, mL	Protein		Activity		Specific Activity, U/mg	Purification Factor, Fold	Yield, %
		mg	%	Units	%			
Supernatant	1470	292	100	138,181	100	473.2	1	100
Ultraconcentrate	5	110	37.7	79,040	57.2	717.9	1.5	57.2
DEAE-Sepharose	28	6.5	2.2	71,025	51.4	10,910	23.1	51.4

**Table 2 biotech-12-00023-t002:** Influence of some metal ions and surfactants on the lipase activity.

Metal Ions (5 mM)	Relative Activity (%)	Detergents, Inhibitors (5 mM)	Relative Activity (%)
Control	100	Control	100
Ca^2+^	106.7	EDTA disodium salt	0
Mg^2+^	93.3	SDS	4.5
Co^2+^	93.3	Tween 20	2.9
K^+^	89.6	DTT	3.3
Na^+^	80.0	PMSF	2.5
Cu^2+^	80.0	NaLS	4.7
Mn^2+^	80.0	NBS	2.8
Hg^2+^	66.7		
Zn^2+^	66.7		
Fe^3+^	33.3

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
