# Peer review of "A Thermostable Lipase Isolated from Brevibacillus thermoruber Strain 7 Degrades Ɛ-Polycaprolactone"

_biotech, 2023, doi:10.3390/biotech12010023_

Round 1
Reviewer 1 Report
Overall, the work is interesting. However, there are few issues that need to be addressed. First of all, there is problem with the title of this work. Authors wrote: “Ɛ-Polycaprolactone degradation…”, however they are discussing isolation of the lipase and corresponding biodegradation. There are two possible outcomes: first is to correct title by adding bio- and provide more data on biodegradation (not only isolation); and the second one to provide more data on physical and chemical properties of the PCL and to proceed with the degradation of the PCL (for example thermal). Likewise, the language and grammar is not at expected level, please enhance it. Keywords – needs to be correlated to the “new” title and the results from this work; use shorter keywords. Do not use term degradation, use biodegradation. By looking Materials and Methods section, one question arises: why other techniques like Infrared Spectroscopy (IR) or Differential Scanning Calorimetry (DSC) were not used? After all, the one of the goals of this work was to determined physical and chemical properties of the PCL or biodegradation products. DSC is very helpful in determination of the melting point or thermal properties of the PCL (page 10, line 294), there is no need to use literature data. IR could confirm assumptions about biodegradation products. Since there are many different techniques, authors are advised to divide Results and Discussion to the corresponding subtitles, like they did in the Materials and Methods section. It would be easier for the readers to follow. Figures 1, 2, 5 and 6 are little vague; resolution needs to be improved. Page 8, line 259: instead of 12-mer please write dodecamer. The main remark of the Discussion section of this work is that reader can gain impression that authors are only confirming or verifying facts already proven in the literature. They need to highlight their results and corresponding conclusions. In that way the novelty of this work will be shown. On the page 10, lines 293-298 authors are discussing melting point of PCL and effectiveness of the enzyme attack. Likewise, they are mentioning enzymatic hydrolytic reactions. Does the moisture have any effect on the (enzyme) biodegradation of the PCL in this work?
Author Response
BULGARIAN ACADEMY OF SCIENCES
INSTITUTE OF MICROBIOLOGY
Acad. G. Bonchev Str., Bl. 26
1113 Sofia, Bulgaria
March 4, 2023
Response to Reviewer 1 Comments
We have modified the manuscript biotech-2256348, " A thermostable lipase isolated from Brevibacillus thermoruber strain 7 able to degrade Ɛ-Polycaprolactone", according to the reviewer comments. All changes are marked in the manuscript in red. Please see below the answers to the reviewer’s notes.
- First of all, there is problem with the title of this work. Authors wrote: “Ɛ-Polycaprolactone degradation…”, however they are discussing isolation of the lipase and corresponding biodegradation. There are two possible outcomes: first is to correct title by adding bio- and provide more data on biodegradation (not only isolation); and the second one to provide more data on physical and chemical properties of the PCL and to proceed with the degradation of the PCL (for example thermal).
Response 1: Title was changed the next way “A thermostable lipase isolated from Brevibacillus thermoruber strain 7 able to degrade Ɛ-Polycaprolactone”
- Likewise, the language and grammar is not at expected level, please enhance it.
Response 2: English revision was made by a native English-speaking colleague.
- Keywords – needs to be correlated to the “new” title and the results from this work; use shorter keywords. Do not use term degradation, use biodegradation.
Response 3: Keywords were changed according to the reviewer’s recommendation.
Keywords: thermostable lipase; enzyme decomposition of plastics; Ɛ-polycaprolactone biodegradation; polycaprolactone biodegradation products
- By looking Materials and Methods section, one question arises: why other techniques like Infrared Spectroscopy (IR) or Differential Scanning Calorimetry (DSC) were not used? After all, the one of the goals of this work was to determined physical and chemical properties of the PCL or biodegradation products. DSC is very helpful in determination of the melting point or thermal properties of the PCL (page 10, line 294), there is no need to use literature data. IR could confirm assumptions about biodegradation products. Since there are many different techniques,
Response 4:
The melting temperature of Ɛ-Polycaprolactone was experimentally determined when we have prepared our previous work (Atanasova, N.; Paunova-Krasteva, T.; Stoitsova, S.; Radchenkova, N.; Boyadzhieva, I.; Petrov, K.; Kambourova, M. Degradation of poly (ε-caprolactone) by a thermophilic community and Brevibacillus thermoruber strain 7 isolated from Bulgarian hot spring. Biomolecules 2021, 11, 1488; doi:10.3390/biom11101488) in aim to determine the temperature for cultivation of the strain. We established that melting temperature is 57°C for Mw 14000 PCL and 60°C for Mw 80000. These data were included in the current version by the next sentence.
“We also established experimentally that the melting temperature was 57°C for Mw 14000 PCL and 60°C for Mw 80000 (unpublished results).”
- authors are advised to divide Results and Discussion to the corresponding subtitles, like they did in the Materials and Methods section. It would be easier for the readers to follow.
Response 5: Results section was divided to four subtitles. As they should be the same in the Discussion section we omitted they there.
- Figures 1, 2, 5 and 6 are little vague; resolution needs to be improved.
Response 6: We have done our best to improve the quality.
- Page 8, line 259: instead of 12-mer please write dodecamer.
Response 7: The change was made.
- The main remark of the Discussion section of this work is that reader can gain impression that authors are only confirming or verifying facts already proven in the literature. They need to highlight their results and corresponding conclusions. In that way the novelty of this work will be shown.
Response 8: The next sentence was included in the Discussion text: “Investigations on the lipase properties demonstrated typical for this subgroup of thermophilic enzymes characteristics, however, the enzyme from B. thermoruber 7 is unique in its ability to degrade PCL.”
- On the page 10, lines 293-298 authors are discussing melting point of PCL and effectiveness of the enzyme attack. Likewise, they are mentioning enzymatic hydrolytic reactions. Does the moisture have any effect on the (enzyme) biodegradation of the PCL in this work?
Response 9: Mentioning enzymatic hydrolytic reactions the authors have had in mind that lipases are hydrolytic enzymes that split off ester bound including a molecule of water.

Reviewer 2 Report
1. Please intensify the significance of degrading PCL at high temperature in the introduction, Why do the polymer degrade by enzyme at high temperature?
2.What is the mean of the sentence in line 338 “a reverse reaction could be suggested after longer time of enzyme action” ?
3. The picture of molecular mass markers in Figure1 is not very clear.
4.The line number covered the Table2, please modify them.
Author Response
BULGARIAN ACADEMY OF SCIENCES
INSTITUTE OF MICROBIOLOGY
Acad. G. Bonchev Str., Bl. 26
1113 Sofia, Bulgaria
March 4, 2023
Response to Reviewer 2 Comments
- Please intensify the significance of degrading PCL at high temperature in the introduction Why do the polymer degrade by enzyme at high temperature?
Response 1: Next sentence was added:
“Thermophilic enzymes able to degrade PCL could be especially useful in this plastic degradation due to its low melting point of around 60 °C.
- What is the mean of the sentence in line 338 “a reverse reaction could be suggested after longer time of enzyme action” ? “
Response 2: The sentence was changed the next way. Hope, now it is more clear:
As the only peak observed in the supernatant after 48 h of a strain cultivation was the monomer [1], the appearance of additional intermediates after longer time of enzyme action could suggested a slow running reverse reaction.
- The picture of molecular mass markers in Figure1 is not very clear.
Response 3: We tried our best to improve it.
- The line number covered the Table2, please modify them.
Response 4: Done
